# Effect of Different Gas-Stunning Conditions on Heme Pigment Solutions and on the Color of Blood, Meat, and Small Intestine of Rabbits

**DOI:** 10.3390/ani12223155

**Published:** 2022-11-15

**Authors:** Muhammad Shahbubul Alam, Dong-Heon Song, Jeong-Ah Lee, Van-Ba Hoa, Hyoun-Wook Kim, Sun-Moon Kang, Soo-Hyun Cho, Inho Hwang, Kuk-Hwan Seol

**Affiliations:** 1Animal Products Utilization Division, National Institute of Animal Science, Rural Development Administration, Wanju-gun 55365, Republic of Korea; 2Department of Animal Science, Jeonbuk National University, Jeonju 561756, Republic of Korea; 3Department of Animal Resources Science, Kongju National University, Yesan 32439, Republic of Korea

**Keywords:** rabbit, inert gases, heme-pigment solutions, carbon-dioxide stunning, halal method

## Abstract

**Simple Summary:**

Ensuring “animal welfare” and “customer satisfaction” are crucial concerns in the modern slaughterhouse. Addressing “animal welfare” includes avoiding or minimizing pre-slaughter stress as well as anxiety, pain, distress, and suffering at the time of slaughter. In general, different methods of gas stunning are used, but the conventional carbon dioxide (CO_2_) gas-stunning method can result in severe aversion, elevated respiratory distress, irritation of the nasal mucous membranes, hypercapnia, and lactic acidosis. To reduce these problems, sometimes inert gases (argon or nitrogen) are mixed with CO_2_ in different proportions. In stunning, it is a vital point that after the onset of unconsciousness, the animal never recovers consciousness. On the other hand, consumers prefer bright red-colored fresh meat. This demands consideration during the stunning and slaughtering of animals. However, different high-concentration levels of inert gases and their effects on stunning are still unknown. To support this, a trial was conducted on heme-pigment solutions (hemoglobin and myoglobin solution) in different high-concentration (80%, 85%, and 90%) levels of CO_2,_ Ar, and N_2_ gas. In both cases, dark red-colored (high redness and low lightness, yellowness) solutions were found in CO_2_-exposed conditions, and bright red was found with N_2_ (high lightness, yellowness, and low redness). The effect of Ar was intermediate between CO_2_- and N_2_-treated groups. These results and concepts were applied to a rabbit-stunning method. It was proven that the rabbit started the onset of unconsciousness at a 96% concentration of N_2_ gas; thus, 96% N_2_ and 98% N_2_ were considered in the nitrogen-treated group. In addition, from every treatment group, 80% concentration levels were omitted. As per the results, dark red-colored blood, meat, and small intestine (high redness and low lightness, yellowness) were seen in CO_2_- and Ar-stunned rabbits, and a bright red color was found in N_2_-stunned rabbits (high lightness, yellowness, and low redness). However, consumers prefer to purchase bright red-colored fresh meat, so N_2_ stunning can help to satisfy their desire. Out of the two treatments, with N_2_, in view of the time required for stunning and color attributes, 98% N_2_ is better for rabbits. Therefore, 98% N_2_ can be considered a better option for gas stunning.

**Abstract:**

The objectives of this study were to elucidate the effect of different high-concentration levels of inert gases (Ar and N_2_) on heme-pigment solutions and on the color of the blood, meat, and small intestine of rabbits; and to figure out the proper concentration level of inert gas (Ar or N_2_) for the complete stunning of the rabbit. To observe the changing of color attributes, a research study was conducted in the abattoir of the National Institute of Animal Science (NIAS), RDA, Republic of Korea. This experiment had 2 parts, (i) A trial on heme-pigment solutions (hemoglobin and myoglobin solution) was conducted in a gas chamber with different high-concentration levels of carbon dioxide, argon, nitrogen, and normal air; and (ii) a rabbit experiment was conducted—stunning with different high-concentration levels of carbon dioxide, argon, nitrogen, and the Halal method (non-stunning). A small-size digital gas chamber was used for this experiment (size: width 850 mm × depth 1350 mm). Artificial hemoglobin (Hb) and myoglobin (Mb) solutions were created from “porcine hemoglobin lyophilized powder” and “equine skeletal muscle myoglobin lyophilized powder”, respectively. In the heme-pigment solutions trial, 10 treatments were used: (i) 80% carbon dioxide (T1), (ii) 85% carbon dioxide (T2), (iii) 90% carbon dioxide (T3), (iv) 80% argon (T4), (v) 85% argon (T5), (vi) 90% argon (T6), (vii) 80% nitogen (T7), (viii) 85% nitogen (T8), (ix) 90% nitogen (T9), and (x) normal air (T10). Heme-pigment solutions (both Hb and Mb) were exposed with each treatment for four separate durations of time (30 s, 1 min, 2 min, and 4 min); and every sample (Hb and Mb) was exposed during each duration of time for 10 times (n = 10). In the rabbit experiment, seven treatments were used (i) 85% carbon dioxide (T1), (ii) 90% carbon dioxide (T2), (iii) 85% argon (T3), (iv) 90% argon (T4), (v) 96% nitrogen (T5), (vi) 98% nitogen (T6), and (vii) the Halal method (non-stunning) (T7). Forty-two rabbits (mixed-breed) were collected from the nearest commercial farm and randomly selected for a treatment group (n = 6). The average body weight was 2.3 kg. For stunning, each individual rabbit was separately kept in a gas chamber, after which each specific gas was used to fulfill the desired level in the pit. After reaching the desired level of concentration, rabbits were stunned within a very short time. It was observed that the time required for stunning in the T1, T2, T3, T4, T5 and T6 treatment was 79–82, 68–73, 97–103, 88–91, 164–173, and 108–115 s, respectively. In the case of the Halal method (non-stunning), as per the rules of Islam, rabbits were slaughtered without stunning. After slaughtering, in all treatments, the blood, meat, and small intestine of each animal were collected carefully and kept in a cool room in which the temperature was −2 °C, and after 24 h, the color attributes—lightness (L*), redness (a*), and yellowness (b*)—were measured via a Chroma Meter. It was found that in both experiments (trial on heme-pigment solutions and rabbit stunning), the color values (L*, a* and b*) showed a significant difference (*p* < 0.05) among the treatment groups. The CO_2_-treated groups showed high redness (a*) and low lightness (L*) and yellowness (b*), which indicated a dark-red color, and N_2_-treated groups showed high lightness (L*) and yellowness (b*) and low redness (a*), which implied a bright-red color. The effect of the Ar was in between the CO_2_- and N_2_-treated groups. These phenomena were observed both in the heme-pigment solutions (Hb and Mb) and in the blood, meat, and small intestine of the rabbits. N_2_-stunned animals fulfill the fundamental desire of consumers to purchase bright red-colored fresh meat. Therefore, in view of color attributes, consumer satisfaction, and animal welfare, N_2_ gas can be thought of as a valuable alternative to stunning. Considering the time required for complete stunning and desirable color attributes, a 98% concentration of N_2_ is best for rabbit stunning. As such, it could be used as a better option for the gas stunning of animals.

## 1. Introduction

Rabbit meat is one of the important lean meats in human dietetics due to its low fat content [1], low cholesterol [2], very low allergenicity, high digestibility [3], and high unsaturated fatty acids [4]. The quality of rabbit meat depends on breed [5], feeding [6], and transport [7,8], as well as the stunning and slaughter methods [9]. In conventional electrical and mechanical stunning and non-stunning Halal methods, rabbits need to be handled and restrained before stunning. During handling and restraining, most abattoir personnel catch the legs or ears of the rabbit, which is painful and not allowed by Regulation 1099/2009 [10] Alternatively, gas stunning lessens human contact during approaching the animal and reduces pre-slaughter stress [11]. Minor stress during the slaughtering of animals perks up meat quality, which influences an increased economic value [12]. Minimum stress during slaughtering and a lower prevalence of dark, firm, and dry (DFD) and pale, soft, and exudative (PSE) meat [13,14] produces a good-quality product with standard color, texture, and juiciness [15]. 

Exposing animals to a high concentration of carbon dioxide (CO_2_) during stunning creates an hypercapnic condition. Hence, changes occur in blood, with a lower pH level, augmented carbon dioxide partial pressure, lesser oxygen partial pressure, lack of oxygen saturation, and minor bicarbonate concentration. As a consequence, the pH of the cerebrospinal fluid will dwindle, resulting in the animal losing consciousness. While stunning, animals are exposed to a 70–98% concentration of CO_2_ and/or a mixture of CO_2_ with N_2_ [16,17]. However, aversion was induced in different species of animals, such as mice, rats, and pigs, by using carbon dioxide-based stunning [18,19,20,21,22,23]. Inhaling a high concentration of CO_2_ results in the irritation of nasal mucosal membranes and lungs in rats [24] and pigs [21]. Severe respiratory distress [23] and strong head shaking [25] were also observed in pigs exposed to a high concentration of CO_2_ during stunning. According to Llonch et al. [26], nasal annoyance and head shaking in rabbits could also be indicative of aversion in carbon- dioxide-based stunning. 

On the other hand, inert gases such as argon (Ar) or nitrogen (N_2_) have been used in the stunning of animals. When inert gases (Ar or N_2_) and CO_2_ were mixed and used in the stunning of animals, the aversion to CO_2_ decreased. It was observed that combinations of N2 with CO_2_ up to 30% showed less aversion in pigs [17,23]. Hypoxic conditions were induced by the inhalation of inert gases [21,27]. Llonch et al. [26] reported that mixtures of N2 and CO_2_ were still averse to rabbits, and they showed fewer signs of unconsciousness. It is a vital point that after the onset of unconsciousness, animals must never recover consciousness. The effect of different inert gases (Ar, N_2_) on the “complete stunning” of animals is also a very important issue. However, the use of different high-concentration levels of inert gas (Ar or N_2_) in the stunning of rabbits and their effect on “complete stunning” has not yet been studied. Supporting and ensuring this fact, a trial was conducted on heme-pigment solutions (hemoglobin and myoglobin solution) in different high-concentration (80%, 85%, and 90%) levels of inert gases, and after this, the obtained results and concepts were applied in a rabbit experiment (stunning with different high-concentration levels of inert gases).

The objectives of this experiment were: (i) to investigate the effect of different high-concentration levels of inert gases (Ar and N_2_) on heme-pigment solutions and on the color of the blood, meat, and small intestine of rabbits; and (ii) to find out the proper concentration level of inert gas (Ar or N_2_) for the complete stunning of the rabbit.

## 2. Materials and Methods

### 2.1. Experimental Design 

To observe the change in color attributes of the blood, meat, and small intestine of rabbits exposed to different high-concentration levels of carbon dioxide, argon, and nitrogen gas stunning, as well as the non-stunning Halal method, a research study was conducted in an abattoir of the National Institute of Animal Science (NIAS), RDA, Republic of Korea. This experiment had 2 parts: (i) a trial on heme-pigment solutions (hemoglobin and myoglobin solution) in a gas chamber with different high-concentration levels of carbon dioxide, argon, and nitrogen gas, as well as normal air; and (ii) a rabbit experiment using stunning with different high-concentration levels of carbon dioxide, argon, and nitrogen gas, as well as the Halal method (non-stunning). A small-size digital gas chamber (width 850 mm × depth 1350 mm) (Supreme Thermal Instrument (STI), Dasa-eup, Dalseong-gun, Daegu, Republic of Korea) was set up in a highly protected and safe room adjoining the animal-slaughtering room. 

### 2.2. Trial on Heme-Pigment Solutions (Hemoglobin and Myoglobin Solution) in Gas Chamber

To conduct this trial, artificial hemoglobin (Hb) and myoglobin (Mb) solutions were created from hemoglobin (porcine) and myoglobin (equine muscle) powder, respectively. A 6-well cell culture plate (flat bottom) was used for this experiment. 

#### 2.2.1. Preparation of Phosphate-Buffered Saline (PBS)

To prepare phosphate-buffered saline (PBS), 1 L de-ionized water was taken in a round-bottom flask. After this, 1 pouch of dry PBS powder (Sigma, St. Louis, MO, USA) was poured in pre-measured distilled water. For proper dissolving, a medium-size magnetic mixer was added into flask. After that, the flask was set in an electric stirring machine for stirring. Around 6 h was required for preparing the phosphate-buffered saline. The yield was 0.01 M phosphate-buffered saline, NaCl-0.138 M, KCl-0.0027 M, pH-7.4. After preparing, the phosphate-buffered saline (PBS) was stored at room temperature (25 °C). 

#### 2.2.2. Preparation of Hemoglobin and Myoglobin Solution

To prepare hemoglobin (Hb) solution, “Hemoglobin porcine lyophilized powder” was collected from Sigma-Aldrich Co., St. Louis, MO, USA. Each vial contained 1 (one) g Hb powder. According to manufacturing guidelines, the solubility of Hb powder in phosphate-buffered saline is 10 mg/mL rate. As per the guidelines for preparing Hb solution, 100 mL PBS was measured by 5 mL pipette and kept in colored bottle. For protection against light entry, the colored bottle was covered with foil paper. After this, a vial of Hb powder was opened very smoothly to avoid any loss of powder. Then, with the help of a digital micro-spoon, Hb powder was taken out from the vial and inserted into the colored bottle in which the pre-measured PBS was kept. After that, the colored bottle was set in an electric stirring machine for stirring. For a proper mixture, a small-size magnetic mixer was added in bottle. For preparing the Hb solution, approximately 2 h time was required for proper mixture. After this, the solution was stored at a temperature of 2–8 °C. 

For preparing myoglobin (Mb) solution, “Myoglobin from equine skeletal muscle, 95–100%, essentially salt-free, lyophilized powder” was also collected from Sigma-Aldrich Co., St. Louis, USA. Each vial contained 250 mg. As per the manufacturing company’s guidelines, first, 25 mL PBS was taken in a colored bottle, and then the entire powder of one vial (250 mg) was inserted with the help of a digital micro-spoon. After that, the bottle was stirred with the help of an electric stirring machine. A small-size magnetic mixer was added into bottle for proper mixture. Approximately 2 h stirring was required for preparing the Mb solution. For protection against light entry, here, the colored bottle was also covered with foil paper. After that, the solution was kept at a temperature of 2–8 °C.

#### 2.2.3. Trial in Gas Chamber

To expose the hemoglobin (Hb) and myoglobin (Mb) solutions in the gas chamber to different high-concentration levels of carbon dioxide, argon, and nitrogen gas, as well as normal air, 10 treatments were used: (i) 80% carbon dioxide (T1), (ii) 85% carbon dioxide (T2), (iii) 90% carbon dioxide (T3), (iv) 80% argon (T4), (v) 85% argon (T5), (vi) 90% argon (T6), (vii) 80% nitrogen (T7), (viii) 85% nitrogen (T8), (ix) 90% nitrogen (T9), and (x) normal air (T10). For obtaining an accurate result, each solution (Hb, Mb) was exposed for every treatment for duration of 30 s, 1 min, 2 min, and 4 min, separately. For each duration (30 s/1 min/2 min/4 min) of every treatment, the sample (Hb or Mb solution) was exposed 10 times (n = 10).

For conducting the trial on the Hb solution, first, a 6-well (flat-bottom) cell culture plate (Thermo Fisher Scientific-KR, Gangnam-gu, Seoul, Republic of Korea) was used. Then, a pre-prepared solution bottle was taken and shaken. After this, 5 mL Hb solution was drawn by pipette and kept in one well of the 6-well plates. The other wells of the plate were also filled up with the same system. On the other hand, the desired gas was supplied in the gas chamber. Gas was supplied from the outer side of the gas cylinder. Gas “in” and “out” of the gas chamber was controlled by a two-flow meter. The concentration of gas in the chamber was detected by a “Gas detector” (New Cosmos Electric Co. Ltd., Seongnam-si, Gyeonggi-do, Republic of Korea). The inner condition of the gas chamber was monitored by an internal close-circuit camera and sensor (Supreme Thermal Instrument (STI), Dasa-eup, Dalseong-gun, Daegu, Republic of Korea), which was connected with a computer on the outer side. For keeping the 6-well plate in the gas chamber, a temporary platform (bench) was placed inside the gas chamber. For obtaining an accurate reading, in the case of carbon dioxide and argon gas treatments groups, a 6-well plate (filled with Hb solution) was inserted in the chamber when the inner concentration of the gas chamber was above 60%. To ensure safety and avoid health hazards, an oxygen mask was used at this time. After inserting the 6-well plate, the cover of the gas chamber was closed quickly and strictly, whereby there was no chance for leakage from the inside. After that, when the concentration of gas reached the desire level, the gas incoming flow meter was stopped and treated at a specific duration of time (30 s/1 min/2 min/4 min). When the specific time was finished, the outgoing flow meter was opened quickly, and the cover of the gas chamber was also opened as soon as possible to take out the 6-well plate (filled with Hb solution). Then, a reading (color values) was taken quickly from every well of the plate with the help of a digital Chroma Meter (Konica Minolta, Tokyo, Japan). One snap (reading) was taken from each well of the plate. Color attributes were denoted as L* (lightness), a* (redness), and b* (yellowness). In the case of the nitrogen gas treatments, first, the 6-well plate (filled with Hb solution) was inserted into the gas chamber, and then gas flow was run, because the stability of nitrogen is slightly lower than air. In the normal air group, the 6-well plate was kept in open air, and a reading was taken after the completion of specific duration of time.

For the trial of the Mb solution, the above 10 treatments were used. Each treatment group was also exposed separately for 30 s, 1 min, 2 min, and 4 min. Ten replications were made (6-well plate was filled with new Hb solution, inserted into the gas chamber, and a reading was taken) for each treatment group and each duration of time (30 s/1 min/2 min/4 min) (n = 10). The same treatments, durations of time, replications, and methods were applied both in the Hb and Mb solution trials. The total duration of this experiment was eight weeks. 

### 2.3. A Rabbit Experiment–Stunning with Different High-Concentration Levels of Gases

#### 2.3.1. Animal Design and Facilities

A very important point to mention here is that the rabbit was not stunned at 85% and 90% concentrations of N_2_ gas. Stunning was started just after the arrival of a 96% concentration. Therefore, some amendments were made in the rabbit experiment: (a) two new treatments were added (96% nitrogen and 98% nitrogen) to replace the 85% nitrogen and 90% nitrogen (those were used in the heme-pigment trial); and (b) for every gas (CO_2_, Ar, and N_2_), the 80% concentration level was omitted. Thus, 7 treatments were used in the rabbit experiment: (i) 85% carbon dioxide (T1), (ii) 90% carbon dioxide (T2), (iii) 85% argon (T3), (iv) 90% argon (T4), (v) 96% nitrogen (T5), (vi) 98% nitogen (T6), and (vii) the Halal method (non-stunning) (T7). Forty-two rabbits (mixed-breed) were collected from the nearest commercial farm. Rabbits were randomly selected for each treatment group (n = 6). The average body weight was 2.3 kg. Rabbits were collected just before 12 h of stunning. From the farm to the abattoir, rabbits were transported by a vehicle of the NIAS. During transport, rabbits were carried in a paper carton and given drinking water. After arrival, rabbits were unloaded and kept in a wire house adjacent to the gas chamber. No feed was supplied in the wire house; only fresh, clean drinking water was supplied for rabbits to drink at an ad-libitrum amount. The staff of the abattoir helped with all of the above. Every day, six rabbits were collected (twice a week). The total duration of the rabbit experiment was four weeks.

#### 2.3.2. Stunning, Slaughter, and Sample Collection

On the morning of the slaughter day (around 9.00 am), the body weight of each rabbit was taken by digital weighing balance and recorded. After this, an individual rabbit was kept in a steel box (length 400 mm × width 400 mm × height 400 mm). One wall of the steel box was transparent (made by transparent fiber glass) for the passage of light, and the other three walls had numerous round holes (for gas entry). Then, the door of box was closed tightly. The rabbit-containing steel box was then shifted from the wire house to the gas chamber room via a trolley and inserted into the gas chamber. The abattoir staffs helped with this. After that, the cover of the gas chamber was closed tightly. Then, the flow meter of a specific gas (Co_2_/Ar/N_2_) was opened, and gas started to flow towards the chamber. All gases were supplied from the outer side of the cylinder. Gas “in and out” of the chamber was controlled by a two-flow meter. The gas concentration of the chamber was observed by a “Gas detector”. For checking the concentration of each gas, a separate “Gas detector” was used. Animals’ movements, behavior, attitudes, and other conditions were monitored by an inner closed-circuit camera and the sensor of the gas chamber, which was connected by a computer on the outer side. The flow meter of a specific gas was stopped after arrival at the desired level of concentration. It was observed that the time required to reach the desired levels of concentration in the gas chamber for T1, T2, T3, T4, T5, and T6 treatments was 7, 10, 12, 16, 27, and 34 min, respectively. After reaching the desired concentration level of a specific gas, the incoming flow meter was stopped, and the time count for stunning was started via stopwatch.The times required for stunning in the T1, T2, T3, T4, T5, and T6 treatments groups were 79–82, 68–73, 97–103, 88–91, 164–173, and 108–115 s, respectively.

After stunning the rabbit, the outgoing gas flow meter was opened quickly for gas to “Go Out”, and the cover of the gas chamber was opened as soon as possible to take out the rabbit-containing steel box. Then, the door of the steel box was opened to take out the stunned rabbit and quickly slaughter it by a sharp knife. During slaughtering, the jugular vein was cut for proper bleeding. From each of animal, 50 mL blood was collected in a conical tube and taken for further collection of results. After this, the head was cut and detached from the carcass. Then, the carcass was skinned. After skinning, loin muscles ware cut carefully and kept in a plastic bag. From every individual rabbit, 100 cm of the small intestine was collected, then cleaned and washed with tap water and inserted in a plastic bag. After this, loin and small intestine samples were stored in a cold room at −2 °C. The remaining carcass was then kept in a plastic bag for eradication. For each treatment group, six animals (n = 6) and thus six small intestines (n = 6) were used.

#### 2.3.3. Color Attributes of Blood, Meat, and Small Intestine

Just after the collection of blood (before coagulation), it was poured into the wells of a six-well round-bottom plate. Then, from every well of the plate, one snap (reading) was taken via a digital Chroma Meter (Konica Minolta, Tokyo, Japan). After 24 h, loin and small intestine samples were taken from the cold room and kept at room temperature (25 °C). A steak of the loin (10 cm × 3.0 cm) was made from each individual and kept in a tray for 40 min for blooming. After full blooming, a reading (snap) was taken from six different locations of each loin sample via the Chroma Meter. In the case of the small intestine, first, a sample (100 cm) was placed longitudinally upon a plastic white board for blooming at 40 min. Then, a reading was taken from 6 different places of the small intestine, and a minimum distance of 10 cm was maintained from one place to another. Before taking the reading, the Chroma Meter was calibrated by a white plate in which Y = 86.3, X = 0.3165, and y = 0.3242. Color values were articulated as L* (lightness), a* (redness), and b* (yellowness).

### 2.4. Statistical Analysis

For data analysis, we used the Statistical Analysis System (SAS) package (Cary, NC, USA, 2007). Means, standard errors of mean (SEM), and *p*-values were calculated for all the treatment groups of both experiments. Standard deviation (SD) was calculated during the Hb and Mb solutions trial. Duncan’s multiple range test was used in both parts of the experiment. The significant difference of both experiments was denoted at *p* < 0.05.

## 3. Results

### 3.1. Effect of Gas Exposed on Hemoglobin Solution

Color attributes of the hemoglobin (Hb) solution that was exposed in high-concentration levels of different gases are presented in Table 1. The results show that all color values (L*, a*, b*) showed a significant difference (*p* < 0.05) among the treatment groups. In L*, the highest value was seen in the T10 group (normal air), and among the gas treatment groups, the highest value was found in the T7 group (N_2_ 80%), and it was lowest in T3 (CO_2_-90%). It was observed that the L* values of CO_2_ gas-treated groups were comparatively lower. Higher values were found in N_2_ groups. Ar gas-treated values were in the middle between N_2_ and CO_2_ groups. It was also observed due to increasing concentration levels of specific gases that the L* value of the hemoglobin solution gradually decreased, such as in the values of the CO_2_ gas-treated T1 and T3 groups. For a*, the highest value was found in the T3 group. CO_2_-treated groups showed a high value of a*, and N_2_-treated groups showed a low value. The color intensity of a* developed while increasing the concentration level of gases. For example, T3 showed a higher value than T1. In b*, N_2_-treated groups showed a high value, and CO_2_-treated groups showed a low value, and values decreased gradually when the concentration level of that gas was extended.

Figure 1A,C indicates that N_2_ treated groups showed high value and CO_2_ treated groups low both in L* and b* of hemoglobin solution. It also mentioned that when the duration of gas exposure time was extended (from 30 s to 4 min), the color value of L* and b* were gradually decreased. Figure 1B illustrates that a* values were developed gradually due to extension of exposure time. High values were observed in CO_2_-treated groups.

### 3.2. Effect Gas Exposed on Myoglobin Solution

Different gas exposure conditions on the myoglobin solution (Mb) are shown in Table 2. The results show that all color parameters (L*, a*, b*) of the myoglobin solution were significantly dissimilar (*p* < 0.05) in the middle of the treatment groups. Elevated L* and b* values were found in N_2_-treated groups (T7 to T9) and were lesser in CO_2_-treated groups (T1 to T3). In both cases, the highest values were seen in T7 and were lowest in T3 among the all gas treatments, although T10 (normal air) showed a higher value than T7 (N_2_ 80%). On the other hand, a* (redness) showed higher values in CO_2_-treated groups (T1 to T3). Of all gas treatments, T3 (90% CO_2_) exhibited the highest value. N_2_-treated groups provided lower a* values of the Mb solution. It was observed that the color intensity of a* was augmented when the duration of exposure was extended, but the values of L* and b* declined gradually. In all cases, the position of the Ar-treated group was in the middle between the N_2-_ and CO_2_-treated groups.

Figure 2A illustrates Lightness (L*) values of Mb. Elevated L* values were seen in N_2_ treated groups and due to extension of exposing time values were gradually decreased in all treatment groups. Figure 2B shows higher values in CO_2_ treated groups than others group. Color intensity was increased while exposing time was extended. Figure 2C indicates that N_2_ treated groups showed high value and CO_2_ treated groups low Yellowness (b*) of myoglobin solution. 

### 3.3. Effect of Gas Stunning on Rabbit’s Blood

The effect of high-concentration levels of different gas-stunning conditions on rabbit’s blood are illustrated in Figure 3. The figure shows that the lightness (L*) value was highest in the Halal method; in the middle of the gas treatment groups, it was elevated in T5 group (96% N_2_). The redness (a*) value was highest in the T2 group. It was observed that among all gas treatment groups, the values of a* were higher in the groups of CO_2_-stunned rabbit’s blood, and L* and b* were comparatively elevated in N_2_-stunned groups. The position of Ar-stunned groups was in the middle. The Halal method showed higher L* and b* values and lower a* values compared to all gas-stunned groups

### 3.4. Effect of Gas Stunning on Rabbit’s Meat and Small Intestine

Table 3 shows the effect of different gas-stunning conditions on the meat and small intestine of rabbits. The results show that among the stunning treatments of different gases at high-concentration levels, noteworthy distinctions (*p* < 0.05) were observed in all the color attributes (L*, a*, b*) of the meat and small intestine of rabbits. In the meat, the lightness (L*) and yellowness (b*) values were higher in N_2_-stunned groups (T5, T6) than for other gases. Redness (a*) was high in CO_2_-treated groups. In the small intestine, the highest redness (redness (a*)) value was found in the T2 group. It is significant that in different methods of gas stunning, elevated a*, and L*, b* values of the small intestine were found in CO_2_- and N_2_-stunned groups, respectively. Ar-treated groups showed medium values between the CO_2_ and N_2_ groups. In both the meat and small intestine, the Halal method provided lower a* and higher L* and b* values than any gas-stunned group.

## 4. Discussion

### 4.1. Gas-Exposed Hemoglobin Solution

The hemoglobin (Hb) molecule is the amalgamation of the globular protein “globin” and a non-protein portion, “heme”. It is composed of four sub-units, each of which contains a heme group and a globin chain. The heme group contains an iron atom in the ferrous form (Fe^2+^) at its center that binds one oxygen molecule, using one hemoglobin tetramer to bind four oxygen molecules [28]. The biological function of Hb is to transport oxygen (O_2_) from the lungs to the muscle cells via the circulatory system, and at the cell wall, it surrenders this O_2_ to myoglobin. Oxygen can also be carried throughout the body by dissolving in blood plasma, but this dissolved portion is a very small portion of the total amount. Only two percent of the oxygen in the bloodstream is dissolved directly in the plasma component of blood compared to 98% of oxygen in the protein-bound state to hemoglobin [29]. Carbon dioxide is transported in the blood from the tissue to the lungs in three ways: dissolution directly into the blood plasma, binding to hemoglobin, or carried as a bicarbonate ion [30]. Carbon dioxide is more soluble in blood than oxygen. According to Henry’s law, the solubility of CO_2_ is 20 times higher than O_2_ at the liquid surface [31]. A maximum of 22% of the carbon dioxide transported by hemoglobin is in the form of “carbaminohemoglobin”, whereas 68% of it is carried as bicarbonate (formed by dissociation of carbonic acid), and 10% of CO_2_ is carried in a dissolved state through plasma [32]. Carbaminohemoglobin is formed by the reaction between carbon dioxide and an amino (NH_2_) residue from the globin molecule, resulting in the formation of a carbamino residue (NH.COO^−^) [33]. In the bicarbonate buffering system, carbon dioxide diffuses into the red blood cells. In the presence of the carbonic anhydrase (CA) enzyme within the red blood cells, carbon dioxide quickly converts into carbonic acid (H_2_CO_3_); and as an unstable intermediate molecule, carbonic acid immediately dissociates into bicarbonate ions (HCO_3_^−^) and hydrogen (H^+^) ions. These H^+^ ions then bind to hemoglobin amino acids [34]. 

When CO_2_ levels are high, the concentration of HCO_3_^−^ and H^+^ ions in the bloodstream is raised too much, lowering the pH and introducing a state of acidosis. The lowered pH of blood then decreases hemoglobin’s affinity for oxygen. Carbaminohemoglobin and H^+^ ions binding to hemoglobin in the amino acid state make it more difficult for O_2_ to bind with hemoglobin [35]. When carbon dioxide binds with hemoglobin, the color changes from bright red to dark red with a hint of purple. Argon gas is heavier than air. The specific gravity of argon and O_2_ gas is 1.39 and 1.113, respectively [36]. Thus, when a high concentration of Ar gas attaches to hemoglobin, it is difficult for O_2_ to combine with it, resulting in the hemoglobin color converting to dark red. On the other hand, nitrogen is lighter than air. Its specific gravity is 0.9737 [36]. Therefore, O_2_ can easily combine with hemoglobin, whereas a high concentration of nitrogen gas (N_2_) attaching with hemoglobin results in the color of hemoglobin being bright red, as in normal blood. In our present study, a high intensity of redness (a*) was observed in the CO_2_-exposed Hb solution, and elevated lightness (L*) and yellowness (b*) were found in N_2_-treated groups. This indicates a dark-red color solution in CO_2_-exposed groups and a bright-red color in the N_2_ group. The color intensity of the Ar-exposed Hb solution was intermediate between the CO_2_ and N_2_ groups.

### 4.2. Gas-Exposed Myoglobin Solution

Myoglobin (Mb) is a protein located primarily in the striated muscles of vertebrates. It consists of a protein component (globin), a prosthetic component, and the ratio of its redox forms in muscle tissue [37]. The main function of Mb is to form an oxygen reserve in muscle tissue, which is consumed during a temporary lack of oxygen [38]. Myoglobin is the reason for the red color of the muscle of most vertebrates [39,40]. Deoxymyoglobin, a reduced form of myoglobin, occurs when no ligand is present at the sixth coordination site and the heme iron is in the ferrous (Fe^2+^) state [41]. Consumer acceptance of the purplish-red color of deoxymyoglobin is low [42]. The deoxymyoglobin state can only be maintained under anoxic conditions, whereby oxygen tensions are very low. When deoxymyoglobin is exposed to oxygen, it allows oxygen to bind to the sixth ligand, producing a bright cherry-red color known as oxymyoglobin. This reaction occurs very quickly, because myoglobin has a high affinity for oxygen and is typically referred to as “blooming” in the meat industry [43]. When myoglobin becomes oxygenated, it still remains in a reduced state (Fe^2+^), while producing a red color preferred by consumers. The oxidation of both oxymyoglobin and deoxymyoglobin leads to metmyoglobin, whereby the heme iron is in the oxidized ferric (Fe^3+^) state and the ligand-binding site is occupied by water [44].

When concentration levels of CO_2_ or Ar are elevated, due to a lack of O_2_ in blood, which lowers pH and decreases hemoglobin’s affinity for oxygen, a very small amount of O_2_ can be exposed to myoglobin, which creates a dark-red color, which supports our present experiment whereby a dark-colored Mb solution (high a* and low L*, b*) was observed in CO_2_- and Ar-treated groups. The intensity of the color change of myoglobin depends on the concentration levels of CO_2_ or Ar. In the case of a lighter gas such as nitrogen, due to a lower specific gravity, a sufficient amount of O_2_ can be exposed to myoglobin and form a bright-red color. The same phenomenon was also found in our study, whereby N_2_-treated groups showed a bright red-colored Mb solution.

### 4.3. Gas-Stunned Rabbit’s Blood

Gas stunning of an animal is accomplished through a neuronal function caused by hypercapnic hypoxia and diminishing pH in the central nervous system. When a high concentration of CO_2_ is used in stunning, *p*CO_2_ in the blood is increased that is subsequently detected by the respiratory center in the medulla oblongata, creating an increased rate of breathing [45]. A huge amount of HCO_3_^−^ and H^+^ ions in the bloodstream was produced in higher levels of CO_2_ stunning. These H^+^ ions bind to hemoglobin amino acids easily and create a barrier for O_2_ to bind with hemoglobin [34]. HCO_3_^−^ may produce an acidic taste and cause irritation to the eyes, nose, and mouth [46]. Irritated mucous membranes include blinking, cleaning of the face, and nasal and eye discharge, which was observed in CO_2_-stunned rabbits [47,48]. Anaerobic oxidative metabolism was increased due to high CO_2_ gas exposure in stunning that raised a large amount of lactate in the bloodstream [49], resulting in lower pH and leading to metabolic acidosis. Unconsciousness is induced through acidosis. Lowering the pH of the blood decreases hemoglobin’s affinity for oxygen. A carbaminohemoglobin state also makes it more difficult for O_2_ to bind with hemoglobin [35]. 

As per the result when carbon dioxide binds with hemoglobin, the color changes from bright red to dark red. This argument was supported by the present experiment, whereby the Hb solution and CO_2_-stunned rabbit’s blood both showed a dark-red color (high a* and low L*, b*). On the other hand, in the case of a high concentration of Ar gas during stunning, a huge amount of gas enters into the body. Ar gas does not directly combine with hemoglobin like CO_2_, but it creates a block for O_2_ to bind with hemoglobin due to a higher specific gravity [36]. Therefore, when lacking O_2,_ blood color observed here also showed a dark-red color, i.e., the same result was displayed in our experiment. In a high concentration of nitrogen gas, O_2_ can bind with hemoglobin, because the specific gravity of nitrogen is lower than O_2_ [36], resulting in a bright-red color of blood. Here, unconsciousness is induced through hypoxia/anoxia.

### 4.4. Gas-Stunned Rabbit’s Meat and Small Intestine

The color attributes of the meat and small intestine is one of the vital points by which consumers evaluate their food’s quality and acceptability. Generally, consumers prefer bright-red fresh meat. Myoglobin is the main heme protein accountable for the red color of meat [50]. When the meat and small intestine are exposed to air for a certain period of time (normally 30 min), myoglobin forms a pigment, oxymyoglobin, which gives a pleasingly cherry-red color. Deoxymyoglobin, a reduced form of myoglobin, results in a purplish-red color, which is not appealing to consumers [42]. The oxidation of both oxymyoglobin and deoxymyoglobin leads to forming metmyoglobin, which creates a brownish color that consumers find unappealing and relate to meat that is no longer fresh [41,51]. The thickness of the oxymyoglobin layer can increase when a sufficient amount of O_2_ is exposed in a certain period of time. It also depends on oxygen’s partial pressure, pH, temperature, and competition for oxygen by other metabolic processes [41]. When inhaling a high amount of CO_2_ in an animal body, due to a lowered pH, carbaminohemoglobin and bonding of H^+^ with hemoglobin, a very small amount of O_2_ can be exposed to myoglobin, as well as large amount of CO_2_ binding with hemoglobin, proving a dark red-colored meat and small intestine. This same phenomenon occurred in our experiment, whereby we found a dark red-colored meat and small intestine (more a* and less L*, b*). These findings agreed with the findings of Onenc and Kaya [52] and Channon et al. [53].

In high concentrations, Ar gas stunning also afforded a dark colored meat and small intestine, because Ar makes a barrier for O_2_ to be exposed in its myoglobin [35]. A contrasting phenomenon was observed in high-concentration nitrogen gas stunning, whereby a sufficient amount of O_2_ can easily bind with hemoglobin and be exposed for long time with myoglobin, resulting in a bright red-colored meat and small intestine [35]. For this same mechanism, bright red-colored meat and small intestine were seen in the present experiment (high L*, b*, and low a*). The color attribute L* passively supports this, with a hue observation along with a*, which is responsible for the 69% inconsistency in the bright-red color [54,55]. The color value a* (redness) is linked with the content of pigment, oxidation situation [54,56,57], and fiber types of the muscle [58]. The intramuscular fat content and redox condition of the meat is related to the color value b* (yellowness) [54,57,59,60].

## 5. Conclusions

As per our findings, N_2_-treated groups showed a bright-red color in pigment solutions, blood, meat, and small intestines of rabbits. A dark-red color was observed in CO_2_-treated groups. Ar-stunned groups provided an intermediate color of CO_2_ and N_2_. When purchasing meat, a bright-red color indicating freshness is a fundamental desire of consumers. N_2_-stunned animals can help to satisfy this desire. Therefore, considering consumer satisfaction, color attributes, and animal welfare, N_2_ can be used as a valuable alternative for gas stunning. In view of the time required for complete stunning and color attributes, a 98% concentration of N_2_ is favorable for rabbit stunning. As such, it could be considered a better option for the stunning of animals.

## Figures and Tables

**Figure 1 animals-12-03155-f001:**
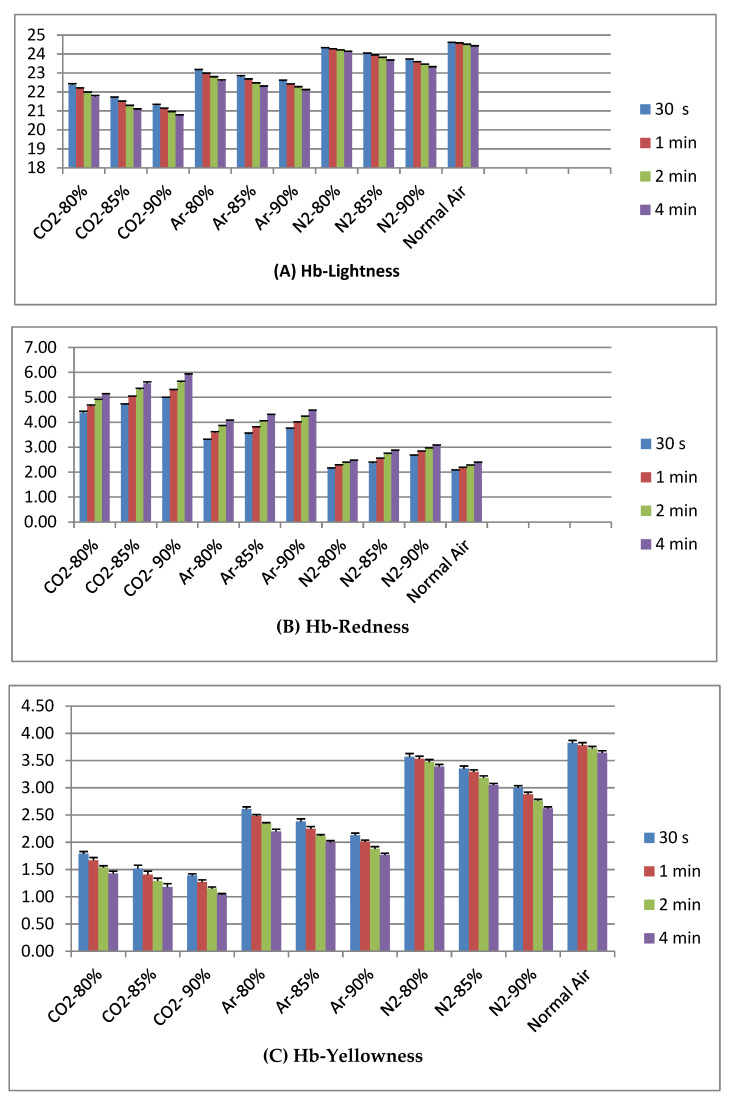
Effect of high concentration levels of different gas exposing conditions on Hemoglobin solution at different duration of time.

**Figure 2 animals-12-03155-f002:**
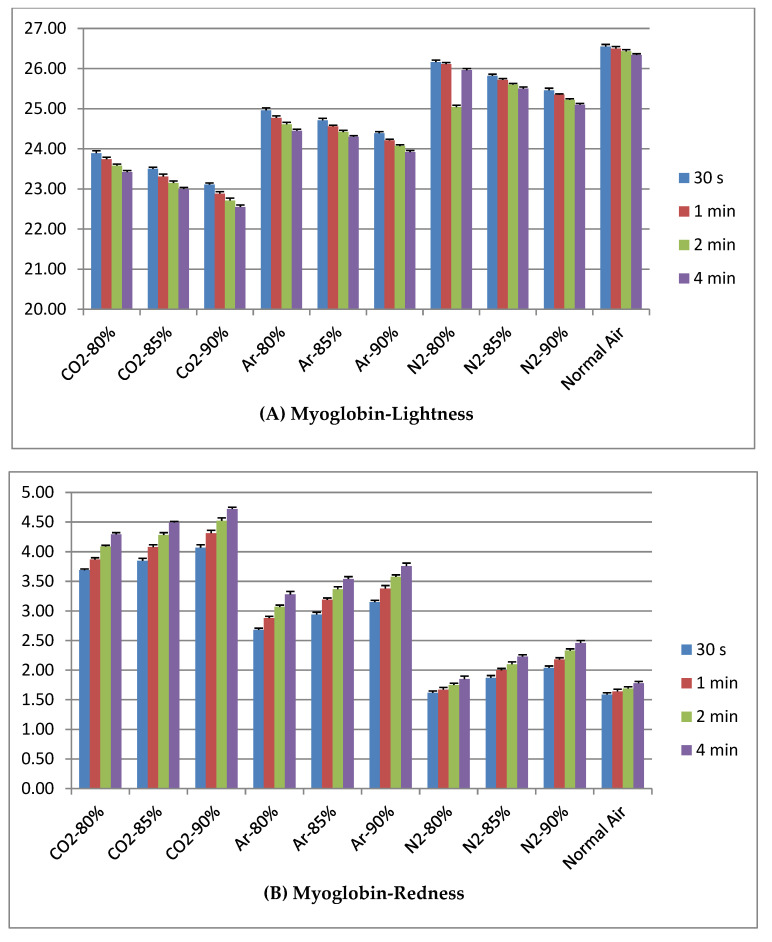
Effect of high concentration levels of different gas exposing conditions on Myoglobin solution at different duration of time.

**Figure 3 animals-12-03155-f003:**
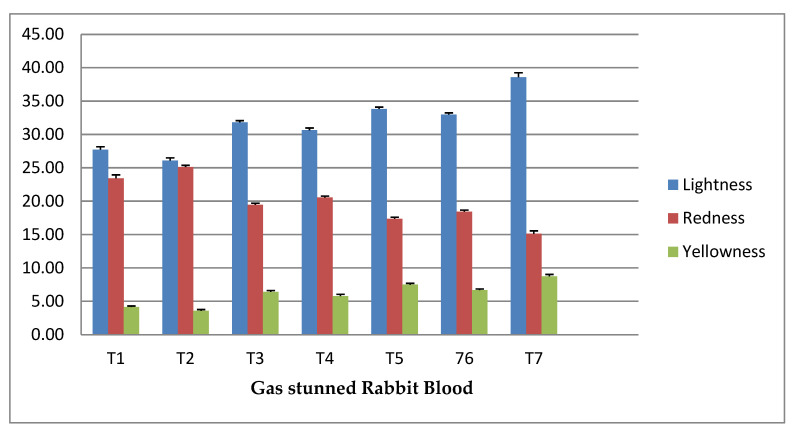
Effect of different gas stunning conditions on Rabbits Blood. T1 = CO_2_-85%, T2 = CO_2_-90%, T3 = Ar-85%, T4 = Ar-90%, T5 = N_2_-96%, T6 = N_2_-98% and T7 = Halal Method.

**Table 1 animals-12-03155-t001:** Effect of high concentration levels of different gas exposing conditions on Hemoglobin solution (n = 10).

Items	Treatments										SEM	*p*-Value
	T1	T2	T3	T4	T5	T6	T7	T8	T9	T10		
Exposing time												
30 s												
L* (Lightness)	22.38^h^	21.69^i^	21.29^j^	23.14^e^	22.80^f^	22.57^g^	24.30^b^	24.00^c^	23.70^d^	24.58^a^	0.105	<0.001
a* (Redness)	4.39^c^	4.70^b^	4.98^a^	3.29^f^	3.53^e^	3.74^d^	2.14^i^	2.36^h^	2.65^g^	2.06^j^	0.102	<0.001
b* (Yellowness)	1.79^h^	1.52^i^	1.39^j^	2.61^e^	2.38^f^	2.13^g^	3.57^b^	3.35^c^	3.00^d^	3.82^a^	0.082	<0.001
1 min												
L* (Lightness)	22.17^h^	21.48^i^	21.10^j^	22.96^e^	22.64^f^	22.39^g^	24.25^b^	23.91^c^	23.57^d^	24.55^a^	0.111	<0.001
a* (Redness)	4.65^c^	5.02^b^	5.28^a^	3.58^f^	3.79^e^	3.98^d^	2.26^i^	2.53^h^	2.82^g^	2.16^j^	0.109	<0.001
b* (Yellowness)	1.67^h^	1.41^i^	1.27^j^	2.48^e^	2.25^f^	2.01^g^	3.53^b^	3.29^c^	2.88^d^	3.78^a^	0.085	<0.001
2 min												
L* (Lightness)	21.96^h^	21.20^i^	20.93^j^	22.76^e^	22.44^f^	22.23^g^	24.19^b^	23.79^c^	23.40^d^	24.49^a^	0.115	<0.001
a* (Redness)	4.89^c^	5.31^b^	5.60^a^	3.84^f^	4.04^e^	4.21^d^	2.37^i^	2.72^h^	2.95^g^	2.27^j^	0.115	<0.001
b* (Yellowness)	1.54^h^	1.29^i^	1.15^j^	2.34^e^	2.12^f^	1.88^g^	3.47^b^	3.18^c^	2.76^d^	3.72^a^	0.087	<0.001
4 min												
L* (Lightness)	21.76^h^	21.06^i^	20.75^j^	22.58^e^	22.27^f^	22.08^g^	24.11^b^	23.65^c^	23.30^d^	24.41^a^	0.119	<0.001
a* (Redness)	5.12^c^	5.56^b^	5.90^a^	4.07^f^	4.29^e^	4.46^d^	2.46^i^	2.85^h^	3.06^g^	2.37^j^	0.122	<0.001
b* (Yellowness)	1.43^h^	1.18^i^	1.04^j^	2.20^e^	2.00^f^	1.77^g^	3.39^b^	3.05^c^	2.62^d^	3.64^a^	0.088	<0.001

T1 = CO_2_-80%, T2 = CO_2_-85%, T3 = CO_2_-90%, T4 = Ar-80%, T5 = Ar-85%, T6 = Ar-90%, T7= N_2_ 80%, T8 = N_2_ 85%, T9 = N_2_ 90%, T10 = Normal Air ^a,b,c,d,e,f,g,h,i,j^ Different superscript letters in same row means significant difference (*p* < 0.05). n = number of times sample exposed in each treatment group. SEM = Standard error of mean.

**Table 2 animals-12-03155-t002:** Effect of high concentration levels of different gas exposing conditions on Myoglobin solution (n = 10).

Items	Treatments										SEM	*p*-Value
	T1	T2	T3	T4	T5	T6	T7	T8	T9	T10		
Exposing time												
30 s												
L* (Lightness)	23.89^h^	23.50^i^	23.11^j^	24.96^e^	24.71^f^	24.39^g^	26.16^b^	25.82^c^	25.46^d^	26.55^a^	0.110	<0.001
a* (Redness)	3.69^c^	3.85^b^	4.07^a^	2.68^f^	2.94^e^	3.15^d^	1.62^i^	1.87^h^	2.04^g^	1.59^i^	0.090	<0.001
b* (Yellowness)	1.93^h^	1.83^i^	1.74^j^	3.17^e^	2.73^f^	2.33^g^	4.27^b^	4.05^c^	3.68^d^	4.43^a^	0.100	<0.001
1 min												
L* (Lightness)	23.74^h^	23.31^i^	22.88^j^	24.77^e^	24.56^f^	24.21^g^	26.11^b^	25.72^c^	25.35^d^	26.50^a^	0.115	<0.001
a* (Redness)	3.87^c^	4.08^b^	4.31^a^	2.88^f^	3.19^e^	3.38^d^	1.67^i^	2.00^h^	2.18^g^	1.64^j^	0.096	<0.001
b* (Yellowness)	1.79^h^	1.68^i^	1.57^j^	3.02^e^	2.59^f^	2.18^g^	4.24^b^	3.95^c^	3.56^d^	4.41^a^	0.104	<0.001
2 min												
L* (Lightness)	23.58^h^	23.15^i^	22.71^j^	24.61^e^	24.42^f^	24.07^g^	26.04^b^	25.60^c^	25.22^d^	26.43^a^	0.118	<0.001
a* (Redness)	4.09^c^	4.28^b^	4.52^a^	3.07^f^	3.37^e^	3.58^d^	1.75^i^	2.10^h^	2.33^g^	1.69^j^	0.101	<0.001
b* (Yellowness)	1.65^h^	1.54^i^	1.42^j^	2.86^e^	2.45^f^	2.03^g^	4.17^b^	3.85^c^	3.43^d^	4.35^a^	0.107	<0.001
4 min												
L* (Lightness)	23.42^h^	23.00^i^	22.55^j^	24.45^e^	24.30^f^	23.92^g^	25.96^b^	25.50^c^	25.10^d^	26.34^a^	0.121	<0.001
a* (Redness)	4.29^c^	4.49^b^	4.72^a^	3.28^f^	3.54^e^	3.76^d^	1.85^i^	2.23^h^	2.46^g^	1.78^j^	0.105	<0.001
b* (Yellowness)	1.53^h^	1.41^i^	1.30^j^	2.72^e^	2.32^f^	1.91^g^	4.09^b^	3.72^c^	3.30^d^	4.28^a^	0.108	<0.001

T1 = CO_2_-80%, T2 = CO_2_-85%, T3 = CO_2_-90%, T4 = Ar-80%, T5 = Ar-85%, T6 = Ar-90%, T7= N_2_ 80%, T8 = N_2_ 85%, T9 = N_2_ 90%, T10 = Normal Air ^a,b,c,d,e,f,g,h,i,j^ Different superscript letters in same row means significant difference (p < 0.05). n = number of times sample exposed in each treatment group. SEM = Standard error of mean.

**Table 3 animals-12-03155-t003:** Effect of high concentration levels of different gas stunning conditions on meat & small intestine of stunned rabbits (n= 6).

Items	Stunning Treatments							SEM	*p*-Value
	T1	T2	T3	T4	T5	T6	T7		
Meat									
L* (Lightness)	49.54^f^	48.43^g^	51.86^d^	51.14^e^	53.43^b^	52.52^c^	56.59^a^	0.388	<0.001
a* (Redness)	3.13^b^	3.34^a^	2.56^d^	2.83^c^	1.94^f^	2.36^e^	1.12^g^	0.111	<0.001
b* (Yellowness)	3.05^f^	2.71^g^	3.63^d^	3.29^e^	4.06^b^	3.83^c^	4.59^a^	0.094	<0.001
Small Intestine									
L* (Lightness)	50.81^f^	48.87^g^	52.76^d^	51.63^e^	55.83^b^	53.94^c^	58.02^a^	0.451	<0.001
a* (Redness)	14.23^b^	16.12^a^	11.85^d^	12.77^c^	10.81^f^	11.28^e^	9.42^g^	0.326	<0.001
b* (Yellowness)	8.74^f^	7.70^g^	11.37^d^	9.82^e^	13.37^b^	12.80^c^	14.07^a^	0.352	<0.001

T1= CO_2_-85%, T2= CO_2_-90%, T3= Ar-85%, T4= Ar-90%, T5= N_2_ 96%, T6= N_2_ 98%, T7=Halal (Non-stunning method). ^a,b,c,d,e,f,g^ Different superscript letters in same row means significant difference (*p* < 0.05). n = Number of animals or small intestines for each treatment group. SEM = Standard error of mean.

## Data Availability

Interested persons can obtain data from the corresponding author on request.

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
