# Peer review of "Effect of Different Gas-Stunning Conditions on Heme Pigment Solutions and on the Color of Blood, Meat, and Small Intestine of Rabbits"

_animals, 2022, doi:10.3390/ani12223155_

Round 1
Reviewer 1 Report
Review for "Effect of Different Gas Stunning Conditions on Heme Pigment Solutions and on the Color of Blood, Meat and Small Intestine of Rabbits"
This manuscript describes a nicely designed and pertinent experiment addressing critical issues in food science. As far as content, design and presentation, I do not see any changes or additions that need to me made. That said, there does need to be extensive English usage and style editing made before publication. Specific defects include the use of symbols (&) for common words which is not appropriate in academic writing, and a general lack of definite articles which are necessary in English. Note that some portions of the manuscript are find (example: most of the abstract), but the lack of language proficiency makes reading the manuscript difficult. I would encourage the authors to have someone go through and fix these issues and resubmit.
Example from the summary:
Original: To ensure ‘animal welfare’ and ‘consumer satisfaction’ is the crucial points in modern slaughter house. Considering animal welfare, all times try to avoid or lessen preslaughter stress as well as slaughter time’s anxiety, pain, distress or sufferings. For this, different gas stunning methods are used. But conventional carbon-dioxide (CO2) gas stunning method has several problems regards animal welfare. Severe aversion, elevated respiratory distress, annoying nasal mucosal membranes, hypercapnia and lactic acidosis are the common feature in CO2 gas stunning.
Corrected English usage: Ensuring "animal welfare" and "customer satisfaction" are crucial concerns in the modern slaughterhouse. Addressing "animal welfare" includes avoiding or minimizing pre-slaughter stress as well as anxiety, pain, distress, and suffering at the time of slaughter. In general, different methods of gas stunning are used, but conventional carbon-dioxide (CO2) gas stunning method can result in severe aversion, elevated respiratory distress, nasal mucosal membrane irritation, hypercapnia and lactic acidosis.
Author Response
Manuscript ID: animals -1966306; Journal - Animals (ISSN 2076-2615)
Manuscript: Effect of Different Gas Stunning Conditions on Heme Pigment Solutions and on the Color of Blood, Meat and Small Intestine of Rabbits
Date: 04.11.22
Dear Reviewer,
Have a nice time. Hope that you are doing well. We appreciate you for your precious time in reviewing our paper and providing valuable comments. The authors have carefully considered the comments and tried our best to address every one of them. We hope the manuscript after careful revisions meet your high standards. The authors welcome further constructive comments if any.
We have provided point-by-point responses in ‘Blue’.
Sincerely yours
Muhammad Shahbubul Alam
First Author
Ph.D Fellow, Jeonbuk National University
& Research Assistant, NIAS, RDA, Korea
Cell: +82-010-4395-0727; email: shahbubulpstu@gmail.com

Reviewer 2 Report
Reviewer comments for manuscript ID animals-1966306 ‘Effect of Different Gas Stunning Conditions on Heme Pigment Solutions and on the Color of Blood, Meat and Small Intestine of Rabbits’
General Comments
It is an innovative study on different gas stunning methods to improve the consumer acceptability of rabbit meat. Improvements in the quality and shelf life of meat is important for the industry in view of the increasing global meat trade. In addition, humane methods of stunning using gas is desirable from the animal welfare point of view and also due to consumer demands and preferences. It is a nice attempt by the researchers to pursue this research. However, I am sorry to say that I struggled to understand the meaning and essence of many sentences in the text due to poor English that might be due to non-native speaking abilities of the authors. I would suggest a professional English editing of the manuscript.
In the assessment of the colour of the meat, how was observer bias eliminated? Please clarify.
Conclusions are very vague, too verbose and do not address the research gaps filled through this study.
I would like to see the corrections/ suggestions done by the authors before I recommend publication of the manuscript.
Specific comments
Lines 12-13: Please reframe ‘To ensure ‘animal welfare’ and ‘consumer satisfaction’ is the crucial points in modern slaughterhouse’ as ‘Consumer satisfaction and animal welfare are critical issue in a modern slaughterhouse’
Lines 13-14: Please reframe ‘Considering animal welfare, all times try to avoid or lessen pre-slaughter stress as well as slaughter time’s anxiety, pain, distress, or sufferings’ as ‘Avoidance of pre-slaughter stress and anxiety, pain, distress and suffering during slaughter is an important animal welfare consideration’
Line 16: Please reframe ‘……. several problems regards animal welfare’ as ‘……. several animal welfare issues’
Lines 16-17: Please rewrite ‘….annoying nasal mucosal membranes’ as ‘ …… irritation of the nasal mucous membranes’
Line 19: Please replace ‘focal point’ with ‘vital’
Line 21: Please reframe ‘This also needs to consider’ as ‘ This demands consideration’
Lines 24-25: Please delete ‘Here findings’
Line 27: Please reframe ‘The position of Ar was middle between CO2 and N2 treated groups’ as ‘The effect of Ar was intermediate of CO2 and N2 treated groups’
Lines 27-28: Please reframe ‘After then, this results and concept was’ as ‘These results and concepts were’
Lines 28-30: Please rewrite the sentences correctly.
Line 36: Please rewrite ‘So it may consider better option of stunning’ as ‘ N2 can be considered as a better option for gas stunning’
Lines 37-73: Abstract – I am sorry to mention, abstract needs to rewritten completely as it is very ambiguous and does not provide a clear picture of the research undertaken.
Line 77: Please reframe ‘content of low fat’ as ‘low fat content’
Line 79: Please replace ‘excellence’ with ‘quality’
Lines 110-12: I am not able to understand these sentences. Please clarify.
Line 133: Please replace ‘which is adjacent nearby’ with ‘adjoining’
Lines 384-96: This is basic biology and well known. Please delete it.
Author Response
Manuscript No. animals -1966306; Journal - Animals (ISSN 2076-2615)
Manuscript: Effect of Different Gas Stunning Conditions on Heme Pigment Solutions and on the Color of Blood, Meat and Small Intestine of Rabbits
Date: 04.11.2022
Dear Reviewer,
Have a nice time. Hope that you are doing well. We appreciate you for your precious time in reviewing our paper and providing valuable comments. The authors have carefully considered the comments and tried our best to address every one of them. We hope the manuscript after careful revisions meet your high standards. The authors welcome further constructive comments if any.
We have provided point-by-point responses in ‘Blue’.
Sincerely yours
Muhammad Shahbubul Alam
First Author
Ph.D Fellow, Chonbuk National University
& Research Assistant, NIAS, RDA, Korea
Cell: +82-010-4395-0727; email: shahbubulpstu@gmail.com

Round 2
Reviewer 2 Report
Reviewer comments on manuscript ID animals-1966306 entitled 'Effect of Different Gas Stunning Conditions on Heme Pigment Solutions and on the Color of Blood, Meat and Small Intestine of Rabbits' Round 2
Abstract : I am sorry the abstract is still not well written. It needs to be written in a proper format.
I am happy with all the other corrections.
Author Response
Manuscript No. animals -1966306; Journal - Animals (ISSN 2076-2615)
Manuscript: Effect of Different Gas Stunning Conditions on Heme Pigment Solutions and on the Color of Blood, Meat and Small Intestine of Rabbits
Date: 10.11.2022
Dear Reviewer,
Good Morning. Hope that you are doing well. Appreciate you for giving precious time in reviewing our paper and providing valuable comments. I have carefully considered your comment and tried my best to write up “Abstract” again as per proper format. For your kind consideration, new Abstract and Manuscript have added as below,
The authors welcome further constructive comments if any.
We have provided point-by-point responses in ‘Blue’.
Sincerely yours
Muhammad Shahbubul Alam
First Author
Ph.D Fellow, Chonbuk National University
& Research Assistant, NIAS, RDA, Korea
Cell: +82-010-4395-0727; email: shahbubulpstu@gmail.com
